# Deformed Mediated Larval Incisor Lobe Development Causes Differing Feeding Behavior between Oriental Armyworm and Fall Armyworm

**DOI:** 10.3390/insects13070594

**Published:** 2022-06-29

**Authors:** Hailong Zhao, Zeng Zhu, Gaoliang Xing, Yiyu Li, Xue Zhou, Jingjing Wang, Guiting Li, Haiqun Cao, Yong Huang

**Affiliations:** 1Anhui Province Key Laboratory of Crop Integrated Pest Management, College of Plant Protection, Anhui Agricultural University, Hefei 230036, China; 18815596956@163.com (H.Z.); a02020b@163.com (Z.Z.); xinggl144148@163.com (G.X.); zhouxuedeyx@sina.com (X.Z.); lgt604@163.com (G.L.); haiquncao@163.com (H.C.); 2Institute of New Rural Development, Anhui Agricultural University, Hefei 230036, China; liyiyu.2012@163.com; 3Plant Protection Station of Anhui Province, Hefei 230061, China; wjj.35@163.com

**Keywords:** *Mythimna separata*, *Spodoptera frugiperda*, mandible, incisor lobe, deformed

## Abstract

**Simple Summary:**

Hox gene *Deformed* (*Dfd*) is important for head appendages, but its function in incisor lobe development is not clear. The different development pattern of incisor lobes resulted in different feeding patterns between oriental armyworm and fall armyworm in maize. The first to sixth instar *S. frugiperda* have sharp incisor lobes, but older instars of *M. separata* have no incisor lobes. Knockdown of *MsDfd* resulted in malformed mandibles with no incisor lobe in *M. separata*, making the larvae unable to perform window-feeding. However, RNAi of *SfDfd* did not affect the mandibles and window-feeding pattern of *S. frugiperda*. Moreover, the mortality of the new first instar *M. separata* increased after feeding with ds*MsDfd* but did not for *S. frugiperda* fed ds*SfDfd*. The results reveal that *Dfd* mediated the larval mandibular incisor lobe morphology, affecting its feeding pattern in *M. separata*.

**Abstract:**

Mandibular incisor lobes are important for insect feeding behavior, living habits and niche. However, the molecular regulation of insect incisor lobe development remains unknown. In this study, we found that two maize pests, oriental armyworm *Mythimna separata* and fall armyworm *Spodoptera frugiperda*, have different feeding patterns in maize, which are closely associated with the different development patterns of their incisor lobes. Different from first to sixth instar *S. frugiperda*, which feed on leaf tissues and whorls with sharp incisor lobes, older instars of *M. separata* feed from leaf margins with no incisor lobes. Hox gene *Deformed* (*Dfd*) is important for head appendages, but its function in incisor lobe development is not clear. Here, *Dfds* were identified from two armyworm species, and both were expressed highly in heads and eggs. Interestingly, the expression levels of *MsDfd* were relatively high in larval mandibles and decreased dramatically from fourth-instar mandibles in *M. separata*. Knockdown of *MsDfd* resulted in malformed mandibles with no incisor lobe in *M. separata*, making the larvae unable to perform window-feeding. However, RNAi of *SfDfd* did not affect the mandibles and window-feeding pattern of *S. frugiperda*, indicating the different roles of *Dfd* in these two species. Moreover, the mortality of new first instar *M. separata* increased after feeding ds*MsDfd* but did not for *S. frugiperda* feeding ds*SfDfd*. These findings revealed that *Dfd* mediated the larval mandibular incisor lobe morphology, affecting its feeding pattern in *M. separata*, broadening the knowledge of *Dfd* functions in insect mandibles and feeding behavior.

## 1. Introduction

Maize *(Zea mays*) is a crop cultivated worldwide with more than 1.14 billion tons in 2018, and China is one of the biggest maize producers and consumers in the world [1,2,3]. Maize faces a serious threat from pests, including two armyworm species, oriental armyworm and fall armyworm, which was first found in January 2019 in China [4]. The oriental armyworm, *Mythimna separata* (Walker), is a polyphagous pest damaging important crops and threatening food security, with the characteristics of a wide range of occurrence, strong reproductive capacity, and long-distance migration [5,6]. The fall armyworm, *Spodoptera frugiperda* (J.E. Smith), damages more than 80 different crops and has spread rapidly in China since 2019, causing serious losses in corn production [7,8,9]. Both armyworm species belong to Noctuoidea, which is the largest and most diverse superfamily of Lepidoptera, with numerous species being important pests [10].

Feeding patterns differ between *M. separata* and *S. frugiperda* larvae in maize seedlings in the field. *S. frugiperda* larvae prefer feeding in the maize whorl, with 89%, 77%, and 100% occupancy rates in V3, V6, and V12 maize, respectively [11]. These larvae feed predominantly, and stay hidden most of their lifetime in maize whorls and stalks in early maize growth stages [12]. By contrast, early instars of *M. separata* feed on leaf tissue and whorls, but older instars feed on leaf margins leaving the leaf midrib behind, which is similar to the armyworm, *Pseudaletia unipuncta* [13]. As Lepidopteran pests, both *M. separata* and *S. frugiperda* larvae have chewing mouthparts. Their maxillae, labium, and hypopharynx degenerate to form a complex, while mandibles are extremely developed and play an important role in feeding. Mandibles, especially their incisor lobes, are necessary for larvae to cut and tear food. The mandible morphology is closely related to the larval feeding behavior and environmental adaptability and is a typical case of the “phenotype-genotype-environment” interaction [10,14].

Mandible development is associated with homeotic genes (Hox), which were first discovered in *Drosophila* and are important transcription factors in insect anterior–posterior body plan, with great significance for analyzing the developmental position and morphology of appendages [15,16]. The anterior Hox genes are important regulators of identity for head appendages, controlling the development of one or more segments and creating a homeotic transformation to the fate of another segment under the removal of each gene [17]. *Deformed* (*Dfd*), a Hox gene, was reported to be highly expressed in the mandibular segments of embryos in *Drosophila* [18,19], *Bombyx mori* [20], *Tribolium castaneum* [21], and *Apis mellifera* [22], and highly expressed in termite soldier mandibles [23]. Moreover, *Dfd* mediated the development of the mandible (node) in insects such as *D. melanogaster*, *Hodotermopsis sjostedti*, and *T. castaneum* [21,23,24]. However, the role of *Dfd* in the development of the mandibular incisor lobe of *M. separata* and *S. frugiperda* larvae and its effect on their feeding behavior are not clear.

This study aims to clarify the difference in larval mandibular incisor lobes between *M. separata* and *S. frugiperda* and their relationships with the different feeding behaviors. Moreover, the function of *Dfd* in the development of incisor lobes and feeding behavior was compared between *M. separata* and *S. frugiperda* larvae.

## 2. Materials and Methods

### 2.1. Insect Cultures

*M. separata* were collected from a field in Yu’an District, Lu’an, Anhui, and *S. frugiperda* were collected from a field in Huangshan District, Huangshan, Anhui [25]. The two species were reared in the laboratory under conditions of 25 ± 1 °C, photoperiod 12 L: 12 D, and 70% relative humidity. First to second instar larvae were fed with fresh corn leaves, third to sixth instars were cultured with an artificial diet, and adults were kept with water containing 10% honey.

### 2.2. Feeding Behavior of M. separata and S. frugiperda Larvae

The feeding positions of 1st–6th instars of the two species were compared on the corn plant in the field. To further analyze the differences in behavior, the 1st–6th instars of two species at the same developmental stage were selected to compare their feeding positions on corn leaves. A transparent plastic device containing 1 larva and 30 larvae were used as a group, with three replicates for each group. A fresh corn leaf of the same size was added into each device, and the initial larval feeding positions on the corn leaves were recorded to judge whether they start eating from the leaf surface or along the edge of the leaf. To analyze the feeding behavior of older instars, larvae were reared on corn leaves, which were sellotaped but not tightly attached to the petri dish.

### 2.3. Morphology of Larval Mandibles

*M. separata* and *S. frugiperda* larvae were selected from 1st–6th instars and dissected in PBS solution. The intact larval mandibles were obtained with scalpels and tweezers under a stereoscope. The morphology of mandibles was pictured using a Nikon camera system (SMZ1500/DS-R12, Nikon, Tokyo, Japan) [26], with ten replicates from each larval stage of the two species.

### 2.4. Characterizations of Dfd from M. feparata and S. frugiperda

The *MsDfd* sequence was isolated from our previous transcriptome dataset [27], and the *SfDfd* sequence was searched from the NCBI database. The open reading frames (ORFs) of *Dfds* were amplified with the templates from the head of 4th instars of two species, using primers designed by Primer premier 5 (Appendix A). PCR amplification conditions were: pre-denaturation at 98 °C for 3 min, 35 cycles of denaturation at 98 °C for 10 s, annealing at 56 °C for 15 s, extension at 72 °C for 10 s, and a final extension at 72 °C for 5 min. PCR products were confirmed by agarose gel electrophoresis and sequencing. Sequence alignment was detected with Clustal Omega (https://www.ebi.ac.uk/Tools/msa/clustalo/, accessed on 1 March 2022). The protein structure and modeling were predicted online by SWISS-MODEL (https://swissmodel.expasy.org/, accessed on 1 March 2022), and the conserved regions were labeled by PyMOL (The PyMOL Molecular Graphics System, Version 2.0 Schrödinger, LLC., New York, NY, USA). The phylogenetic tree was built with MEGA 7.0 [28] using the maximum likelihood (ML) method. Orthologs of Dfd were collected from different insect species from the NCBI database (Appendix A). The branch support of the tree was estimated with ultrafast bootstrapping testing with 1000 replicates.

### 2.5. Spatiotemporal Expression Patterns of MsDfd and SfDfd

Expressions of *Dfds* were compared in different tissues, at different developmental stages, and among the mandibles of 3rd–6th instars of two species, respectively, with three biological replicates for each sample. Different tissues were dissected from 4th instars, including integuments, midguts, Malpighian tubules, fat bodies, ventral nerve cords, brains, and mandibles. Eggs, 1st–6th instar larvae, pupae, and male and female adults at the first day of each stage were collected with three replicates. The RNAs of mandibles were extracted using SPARKeasy Tissue/Cell RNA Rapid Extraction Kit (SparkJade, Jinan, China) and the RNAs of other samples were extracted with Trizol reagent (Thermo Fisher Scientific, San Jose, CA, USA). RNA quality was checked with a Nanodrop ND-1000 spectrophotometer (DeNovix, Wilmington, DE, USA) and confirmed by agarose gel electrophoresis. First-strand cDNA was synthesized with PrimeScript™ RT reagent Kit with gDNA Eraser (Perfect Real Time, Takara, Dalian, China) according to the manufacturer’s protocol. Real-time quantitative PCR (RT-qPCR) assay was performed in a Bio-Rad iCycler iQ Real-time Detection System (Bio-Rad, Hercules, CA, USA) using TB Green Premix Ex Taq II (Tli RNaseH Plus, Takara) following the manufacturer’s manual. The conditions for RT-qPCR reactions were: 95 °C for 2 min, 40 cycles of 95 °C for 15 s and 60 °C for 30 s, with a dissociation curve range from 60 to 95 °C at the end. The internal reference genes of *M. separata* and *S. frugiperda* were *ribosomal protein 49* (*rp49*) and *Ribosomal Protein L18* (*RPL18*), respectively.

### 2.6. Function of Dfd in M. separata and S. frugiperda

Fragments of *MsDfd*, *SfDfd*, and *GFP* were amplified by PCR using primers designed by Primer premier 5, containing the T7 RNA polymerase promoter (Appendix A). Fragments were sequenced and used to synthesize dsRNA using a T7 RiboMAX Express Large-Scale RNA Production System kit (Promega, Beijing, China). dsRNA of *GFP*, *MsDfd*, and *SfDfd* was diluted to 1000 ng/μL, and was injected into the 3rd instar larvae through the IM-31 microinjection system (Narishige, Tokyo, Japan). Each larva was injected with 2 μL of dsRNA, with 30 larvae for each gene. Injected larvae were reared on an artificial diet individually. The knockdown efficiency of *MsDfd* and *SfDfd* was determined using RT-qPCR with three randomly collected larvae from each of the ds*GFP*- and ds*Dfd*-injected groups at 24 h post injection, with three biological replicates, respectively. The phenotypes of mandibles and crochets were observed and photographed, and larval feeding behaviors on corn leaves were compared. Moreover, dsRNA was diluted to 200 ng/μL and sprayed onto corn leaves. Thirty newly hatched larvae were fed on pre-treated corn leaves, with three replicates for each treatment. Larvae fed on ds*GFP*-treated leaves were used as the control. Morphology and survival were recorded every 24 h after feeding dsRNA-treated leaves.

### 2.7. Data Analysis

The SPSS 22.0 software was used to perform all statistical analyses. The difference in spatiotemporal expression of *Dfd* was analyzed by one-way analysis of variance (ANOVA), followed by Duncan’s test with *p* < 0.05. RNAi silencing efficiency and mortalities of larvae fed on dsRNA were tested by independent-samples *t* test with *p* < 0.05. GraphPad Prism 8 software was used for graphing.

## 3. Results

### 3.1. Different Feeding Behavior between M. separata and S. frugiperda Larvae

During the vegetative growth stages of maize in the field, new first instars of *M. separata* and *S. frugiperda* feed on the leaf surface and then move to whorls. Second to third instars of both pests prefer feeding in whorls. By contrast, fourth to sixth instars of *M. separata* move out of the whorl and feed from the leaf edge, but older instars of *S. frugiperda* still feed in the whorl (Figure 1A,B). To analyze why *M. separata* older instar larvae like feeding from the leaf edge, their abilities of biting and boring leaves were tested. The results showed that first to second instars of both *M. separata* and *S. frugiperda* tended to feed from the leaf surface, not from the edge, forming a window-like damage (Figure 1C,D). The feeding behavior of third instars began to differ between the two species. Most of the third to fourth instar *M. separata* started feeding from the leaf edge, while most of the third to fourth instar *S. frugiperda* fed from the leaf surface. Interestingly, there were still some fifth to sixth instar *S. frugiperda* that fed from the leaf surface; however, all fifth to sixth instar *M. separata* could only feed from the leaf edge (Figure 1E,F). To identify the disability of older instar *M. separata* feeding from the leaf surface, sixth-instar larvae were reared on corn leaves which were placed inside on the petri dish. The results showed that sixth-instar *S. frugiperda* fed from the surface, forming a hole in the leaf; however, sixth-instar *M. separata* could not bore the leaf to form a hole (Figure 1G). Therefore, all instars of *S. frugiperda* larvae could feed from the leaf surface, while fifth to sixth instar larvae of *M. separata* could only feed along the leaf edge.

### 3.2. Morphological Differences between M. separata and S. frugiperda Larval Mandibles

Both *M. separata* and *S. frugiperda* larvae have chewing mouthparts with well-developed mandibles. First to third instar *M. separata* have four sharp teeth on the mandible, but fourth to fifth instars have only three or fewer teeth on the mandible. In the last instar of *M. separata*, the mandible bears no teeth but a semi-circular blade and becomes more robust (Figure 2A). By contrast, first to sixth instar *S. frugiperda* have “toothed shape” mandibles with four distinct sharp incisor lobes on the distal edge (Figure 2B).

### 3.3. Characterizations and Expressions of MsDfd and SfDfd

*Dfd* genes were identified and cloned from *M. separata* and *S. frugiperda*, both with an ORF of 1185 bp. The protein sequence alignment showed that MsDfd and SfDfd contained the homeobox domain and shared 98.7% similarities with each other (Figure 3A), having similar predicted protein structures (Figure 3B,C). Phylogenetic analysis showed that MsDfd and SfDfd were clustered into same branch, closely related to other Lepidopteran Dfds (Figure 3D).

RT-qPCR results showed that the expression level of *MsDfd* was significantly higher in the head, ventral nerve cord, brain, and mandible than in the other tissues of fourth-instar *M. separata* (*p* < 0.05) (Figure 4A). *MsDfd* was highly expressed in eggs/embryos (Figure 4B). Among third to sixth instar larval mandibles, the expression level of *MsDfd* was significantly higher in the third instar than fourth to sixth instars (*p* < 0.05, Figure 4C). By contrast, *SfDfd* was highly expressed in the head, but low in the mandible in fourth-instar *S. frugiperda* (Figure 4D). *SfDfd* was highly expressed in eggs/embryos (Figure 4E), and had similar expression levels among third-, fifth-, and sixth-instar larvae (Figure 4F).

### 3.4. Functions of Dfds in Mandible Development and Larval Feeding

RNAi of *Dfds* was performed with dsRNA-microinjection for third-instar larvae. As a result, both *MsDfd* and *SfDfd* were knocked down in third-instar larvae after injecting dsRNA, with a 43% and 75% decrease at 24 h, respectively (Figure 5A,B). The results showed that fourth-instar larvae molted from ds*MsDfd*-injected third-instar *M. separata* had shorter, thinner and flatter mandibles than controls injected with ds*GFP* (Figure 5C), while ds*SfDfd*-injected *S. frugiperda* larvae showed no abnormal mandibles (Figure 5D). For ds*MsDfd*-injected larvae, it was difficult to feed on the leaf surface to form window damage (Figure 5E), but ds*SfDfd*-injected larvae could feed from the leaf surface to form a hole as controls (Figure 5F). Moreover, the effects of dsRNA feeding on newly first instar larvae were compared between these two species. Interestingly, *M. separata* larvae had higher mortalities after feeding with ds*MsDfd* than controls fed ds*GFP* for 24, 48, and 72 h, respectively (Figure 5G), while the mortality of *S. frugiperda* larvae was similar to that of control groups (Figure 5H).

## 4. Discussions

*M. separata* and *S. frugiperda*, which show different feeding behaviors in the field, are two major pests in maize worldwide. Their feeding patterns were compared in maize seedlings in this study. Our results ascertained that older instar *M. separata* fed from leaf edges while older instar *S. frugiperda* larvae preferred feeding on whorls and could chew holes as early instars of both species. *M. separata* is similar to the armyworm, *P. unipuncta*, whose early instars perform window-feeding and older instars perform edge-feeding [13].

Compared to *S. frugiperda*, larval mandibles of *M. separata* exhibited marked ontogenetic changes. The mandibular incisor lobes of *M. separata* larvae began to disappear at the fourth instar, with incisor lobes becoming short and smooth, and even absent and “blade-bearing type” in the last instar, which is similar to *M. sequax*, *Rabtala cristata* and some notodontid larvae [29,30,31]. Different to the armyworm, *P. unipuncta*, with dentate mandibles for first to third instars and chisel-like mandibles for fourth to sixth instars, fourth to fifth instars of *M. separata* still have dentate mandibles [13]. In noctuid larvae, the mandibular incisor lobes are generally considered to be for biting and severing, and the change in lobes will produce a functional alteration in how the larvae cut the plant tissues. A sharp-toothed mandible is effective for newly hatched first instars to feed on the relatively soft mesophyll tissue between the veins, forming “windowpane” feeding damage. The feeding pattern of the fifth to sixth instars of *M. separata* is “snipping type”, cutting from the edge of leaves. The blade-bearing type of mandible could provide a biomechanically considerable bite force strength to deal with even tough leaves [29].

Ontogenetic changes in mandibles were correlated with feeding patterns. Caterpillar mandibles are renewed during molting to provide potentially different feeding strategies during their growth and development. First instars and early second instars are too small and weak to bite through the leaf margin, but the dentate mandibles make them suited to window-feeding despite being unable to sever leaf veins. The dentate mandibles of third to sixthinstars of *S. frugiperda* are suited to window- and edge-feeding, and their larvae prefer whorls and bore maize seedlings. By contrast, with no incisor lobes, older instar *M. separata* with chisel-like edges on mandibles are suited to snipping leaf edges, contributing to their edge-feeding in the field. Morphological variation in functional structures, such as the mandible, can affect the herbivory interaction with crops and indicate an adaptability of the pest to expand and colonize new regions with the consequent economic losses [10]. The relationship between morphology and ecology could provide helpful insights into the expression of the environment–phenotype–genotype interaction.

*Dfd* is critical for the formation and maintenance of the feeding unit required for feeding behavior [32,33]. In strong *Dfd*¯ phenotypes of *Drosophila* larvae, which have a cephalopharyngeal skeleton, the mouth hooks and H-piece were missing [34,35]. *Dfd* contributed to the development of insect mandibles. *Dfd* genes were identified from *M. separata* and *S. frugiperda* in this study, and two *Dfds* were extremely similar in their sequences, containing the YPKM motif and homeodomain [36]. *MsDfd* and *SfDfd* were expressed highly in the larval head, which is similar to *Dfd* in silkworm (SilkDB 3.0) and fruit fly (FlyBase). *MsDfd* had high expression levels in mandibles, such as *NtDfd* in mouthparts in *Nasutitermes takasagonesis*, indicating its function in the development of mandibles [37,38]. From the phenotypes produced by RNAi, *MsDfd* was inferred as the sole gene responsible for mandibular identity in *M. separata*, with the depletion of *Dfd* producing malformed mandibles without sharp incisor lobes, which was consistent with its high expression in larval mandibles. *Dfd* could inhibit mandibular regression, probably resulting from the decrease in dead cells [37]. The expression levels of *MsDfd* dramatically decreased from the fourth instar, accompanied by ontogenetic changes in mandibles from the fourth instar. Thus, *Dfd* is necessary for proper mandibular incisor lobe development in *M. separata*. It is reported that *Dfd* regulated downstream genes and was regulated by upstream genes to perform specific functions in insects [39]. However, *Dfd*-regulated downstream genes, which control the development of mandibles, especially their incisor lobes, need to be identified in further studies. Moreover, the different expression patterns and functions of *Dfd* in *S. frugiperda* probably result from the diversity in the transcriptional regulation of *Dfd* and its regulated downstream genes.

In conclusion, this study showed that the oriental armyworm and fall armyworm damage maize differentially, having different feeding patterns in maize in the field. The anatomy of larval mouthparts showed that ontogenetic changes in mandibles are associated with an alteration in feeding behavior. Moreover, *Dfd* mediated the larval mandibular incisor lobe morphology, affecting its feeding patterns in *M. separata*. These results will broaden our knowledge of *Dfd* function in incisor lobe development in insects and provide novel insights into insect feeding behavior, guiding the scientific usage of pesticides in the field.

## Figures and Tables

**Figure 1 insects-13-00594-f001:**
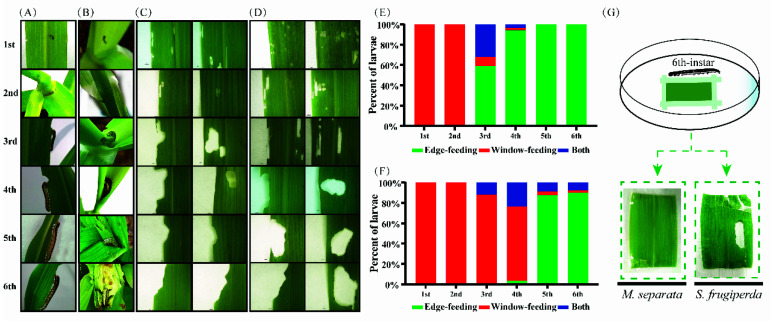
Feeding behaviors of two armyworm species. Feeding patterns of 1st–6th instar oriental armyworm (**A**) and fall armyworm (**B**) in maize in the field. Feeding patterns of 1st–6th instar oriental armyworm on maize leaves, with two leaves representing window-feeding for 1st–2nd instar, edge- and window-feeding for 3rd–4th instar, and edge-feeding for 5th–6th instar larvae (**C**). Feeding patterns of 1st–6th instar fall armyworm on maize leaves, with two leaves representing window-feeding for 1st–2nd instar and edge- and window-feeding for 3rd–6th instar larvae (**D**). Percentages of different feeding patterns of 1st–6th instar oriental armyworm (**E**) and fall armyworm (**F**) on maize leaves. (**G**) Window-feeding abilities of two armyworm species with larvae reared on maize leaves sellotaped on the petri dish.

**Figure 2 insects-13-00594-f002:**
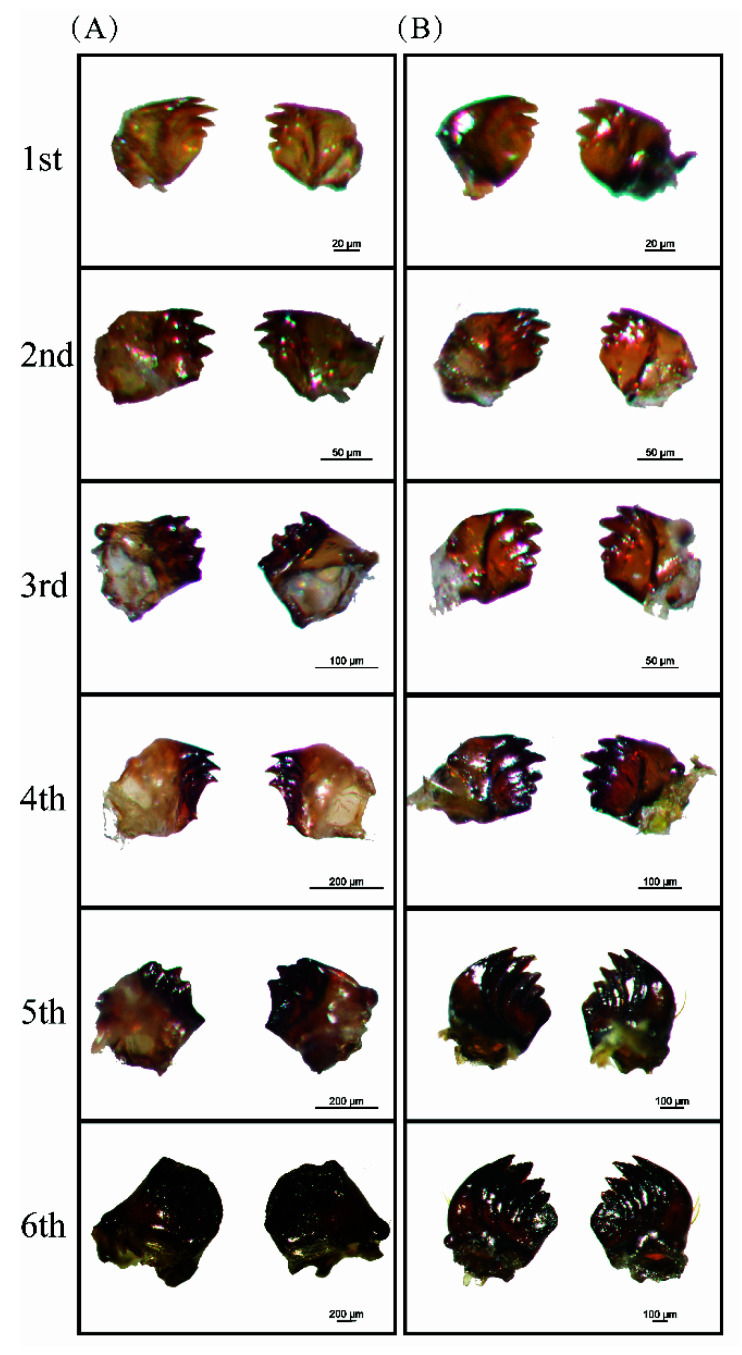
Morphology of larval mandibles of two armyworm species. (**A**) 1st–6th instar oriental armyworm. (**B**) 1st–6th instar fall armyworm. Larval mandibles were dissected from mouthparts and photographed.

**Figure 3 insects-13-00594-f003:**
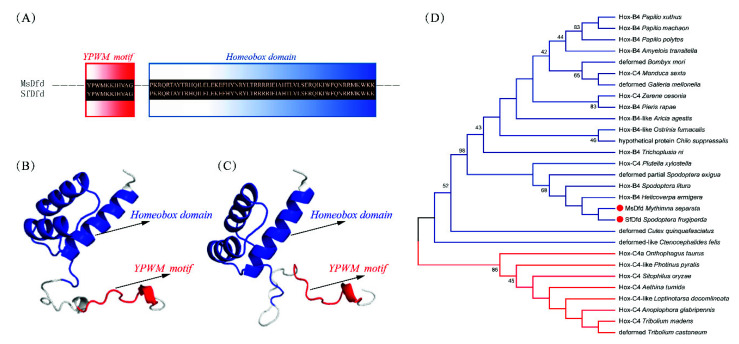
Characterizations of MsDfd and SfDfd. (**A**) YPWM motif and homeobox domain of MsDfd and SfDfd. Predicted protein structures of MsDfd (**B**) and SfDfd (**C**). (**D**) Phylogenetic tree of Dfds. Dfds marked with a red dot are MsDfd and SfDfd. The phylogenetic tree was built using MEGA 7 with the maximum-likelihood method. Red line represents Dfds from Coleopteran insects and blue line represents Dfds of insects from other orders.

**Figure 4 insects-13-00594-f004:**
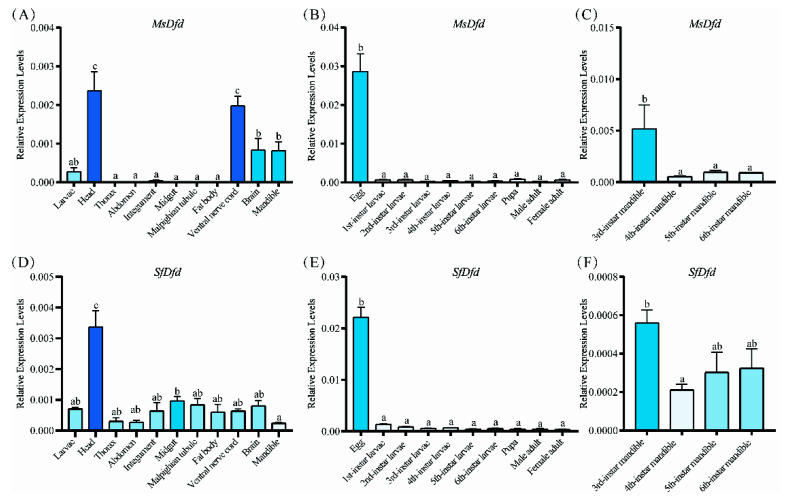
Spatiotemporal expression patterns of *MsDfd* and *SfDfd*. (**A**) *MsDfd* in different tissues of 4th-instar *M. separata*. (**B**) Levels of *MsDfd* during different developmental stages of *M. separata*. (**C**) Expression of *MsDfd* among mandibles of 3rd–6th instar *M. separata*. (**D**) *SfDfd* in different tissues of 4th-instar *S. frugiperda*. (**E**) Levels of *SfDfd* during different developmental stages of *S. frugiperda*. (**F**) Expression of *SfDfd* among mandibles of 3rd–6th instar *S. frugiperda*. The 2^−ΔCt^ method was used to calculate the relative gene expression. Three biological replicates were performed for each group. The bars with different colors and different numbers on the columns represent significantly different levels of gene expression (*p* < 0.05, Duncan’s multiple range test).

**Figure 5 insects-13-00594-f005:**
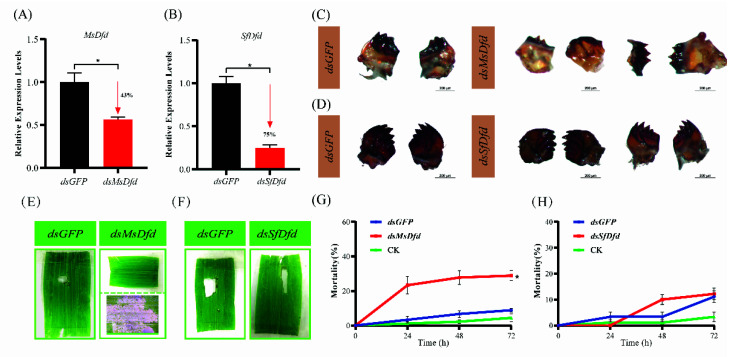
Functions of *Dfds* in larval mandible development and feeding behavior. Knockdown efficacy of *MsDfd* (**A**) and *SfDfd* (**B**). Differences of mRNA levels were analyzed by Student’s *t*-test (*p* < 0.05), with * and red bar showing the significance. The arrow represents the decrease of mRNA levels. Morphological changes of mandibles after knockdown of *MsDfd* in *M. separata* (**C**) and *SfDfd* in *S. frugiperda* (**D**). Larval feeding changes after knockdown of *MsDfd* in *M. separata* (**E**) and *SfDfd* in *S. frugiperda* (**F**). Top panel for *dsMsDfd*: larvae cannot feed on the leaf; Bottom panel for *dsMsDfd*: larvae can only browse mesophyll (**E**,**G**) Mortality of newly first instar *M. separata* feeding on ds*MsDfd*-containing diet. (**H**) Mortality of newly first instar *S. frugiperda* feeding on ds*SfDfd*-containing diet.

## Data Availability

The data presented in this study are available in article and supplementary materials. The raw data used and/or analysed during the present study are available from the corresponding author on reasonable request.

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
