# Peer review of "Deformed Mediated Larval Incisor Lobe Development Causes Differing Feeding Behavior between Oriental Armyworm and Fall Armyworm"

_insects, 2022, doi:10.3390/insects13070594_

Round 1
Reviewer 1 Report
This manuscript revealed the feeding pattern of two maize pest Mythimna separate and Spodoptera frugiperda according to the change of their mandible morphology. The result is interesting to find the function of Dfd to mediate larval mandibular incisor morphology and affect their feeding behavior. There are some minor problem should be revised.
Line41, “Mythimna separate; Spodoptera frugiperda” should be italic.
Line47, “detected” change to “found”
Line49, delete “cash”
Line 57, How to understand “V3, V6 and V12 maize”?
Line79-80, The sentence “However, the function of Dfd…remain unknown. The importance of…is not clear.” Should be rewrited
Line113-115, the source of analytic software should be cited.
Line139, in TableS1, where did these primers come from?
Line178, “the older instar” can change to “5th-6th instar larvae of”
Line187-192 and Figure2, adjusting the order of Figure2A and Figure2B, or change the description in paragraph.
Line204, Figure 3D, the latin name of insect species should be italic.
Line207, “circles” change to “red dot”
Line218, From the result of Figure 4, the level of SfDfd and MsDfd are high in the mandible of 3rd-instar or 3-6th instar in Figure 4C and 4F, but they are very low in the whole larvae stage in Figure 4B and 4E. The mandible is the part of larval head, how to explain this result?
Line245-246, “with different insect herbivory in the field” is not right.
Line 267, “in their feeding life” change to “during their growth and development”
Line 317-319, delete the number label in front of two references.
Line 321, correct the format of the page numbers
Reviewer 2 Report
The manuscript entitled “Deformed mediated larval incisor lobe development differing feeding behavior between oriental armyworm and fall armyworm” analyzed the roles of one hox gene in two armyworm. This study is well designed and analyzed. I have some small concerns and suggestions as below.
Abstract
Line 37, revealed.
Line 41, italic of the Latin name of two species. The full name of the dfd gene in keywords.
Introduction
Line 68, “which were first discovered”.
2.2 section
Line 99, “were recorded”.
2.3 section
Line 107, provide the info. of the device.
2.4 section
How did the PCR perform, and what’s the primers for PCR? in table S1?
2.5 section
The authors should provide the details of the sample collection and preparation, i.e. the day-age of the pupa and female and male adults.
Line 125, “instar larvae”.
Line 126, Jinan, China? The country should be provided of the devices and kits, also line 127,
Line 134, “for” 2 min, “for” 15 s, etc.
3.3 section
Line 199, the ORF is referred to the nucleotide sequence, not the amino acid. Plz confirm.
References
The first two references were double numbered.
Figures
The quality of the figures is not good enough. Better re-organize the figures with high resolution or the vector figures.
The citation of the figures also the panel in each figure must be orderly cited in the text, i.e. Figure 4, Figure 5.
What are the differences of the colored bars in figure 4? The authors should explain in the legends or some where.
Reviewer 3 Report
Comments to authors
The study by Zhao et al. focuses on identifying feeding pattern differences between two closely related Noctuidae family Lepidopteran insects. They showed that Oriental armyworm mouthparts altered during late instar larval stages which eventually affect its feeding behavior on maize. However, they do not see any differences in the feeding behavior of fall armyworm. They claim that Hox gene, Deformed expression determines the morphology of the mandibular incisor lobe in the Oriental armyworm. But it seems to be Deformed is not playing any role in mandibular incisor lobe morphology in the Fall armyworm confirmed by its low expression in mandible and knockdown studies.
This paper provides some good evidence on how closely related species avoid competition for food by changing their mouthparts and feeding behavior. However, I have some queries on this study as listed below.
Authors show that there is no structural difference in the Deformed protein, though its expression pattern varied between these two species. Knockdown of Hox gene-Deformed affected mandible development and reduced the feeding in Oriental armyworm but not in fall armyworm. It is interesting to know why two closely related species have different regulatory mechanisms to control the expression of Deformed and mandible development. Authors can discuss other regulatory mechanisms or genes involved in mandible development (especially in the fall armyworm). This helps in understanding how closely related species evolve morphological differences that are very likely related to the divergence of feeding habits on their special host plants.
Line No. 100-102: I wonder if attaching corn leaves to Petri dishes with sellotape may hinder the larval feeding. It is ideal to place corn leaves in the petri dish without attaching. Did authors observe any differences in feeding when leaves were attached to the petri dish?
Line No. 175: Replace ‘stuck’ with ‘placed inside’.
Fig. 1C &1D: Both of these figures are a bit confusing. Why are the 2 panels in Fig. 1C and Fig. 1D? What authors want to convey is not clear. Each panel needs separate labeling or provide a description in the figure legend.
The feeding bioassay results are consistent in different bioassays? Did authors repeat feeding bioassays independently? No information is available on this. To make sure results are reproducible, authors need to repeat bioassays at least three times.
Line No. 212: Replace ‘lowly’ with ‘low’.
Fig. 3D. Authors need to provide accession numbers for Dfds of different species used for phylogenetic tree construction.
Fig. 4. How many replicates are used for relative gene expression analysis? Did you repeat this experiment independently? Please include this information in the figure legend. Also include a description of statistical analysis in the figure legend.
Lino No. 224: Replace ‘As the results’ with ‘As a result'.
Fig. 5E. Authors need to provide descriptions for the top and bottom panels in the dsDfd treatment.
